# Extensometer for Determining Strains on a Tensile and Torsion Simultaneous Load

**DOI:** 10.3390/s20020385

**Published:** 2020-01-09

**Authors:** Viorel Goanta

**Affiliations:** Mechanical Engineering, Mechatronics and Robotics Department, Mechanical Engineering Faculty, “Gheorghe Asachi” Technical University of Iasi, 700050 Iasi, Romania; vgoanta@tuiasi.ro

**Keywords:** extensometer, tensile strain, shear strain, calibration, elastic deformation, combined loads

## Abstract

The paper presents an extensometer designed to measure two mechanical strains at the same time—one from tensile load and the other from torsion load. Strain transducers provide different electric signals, which, after calibration, lead to the simultaneous measurement of linear (ε) and angular (γ) strains. Each of these two signals depends on the measured process and is not influenced by the other strain process. This extensometer is designed to be easily mounted on the sample with only two mounting points and can be used to measure the combined cyclical fatigue of tensile and torsional loadings. This extensometer has two bars—one rigid, reported at the resulting stress points, and one elastic and deformable. The elastic deformable bar has two beams with different orientations. When the sample is deformed, both beams are loaded by two bending moments (perpendicular to each other and both perpendicular on the longitudinal axis of the bars).

## 1. Strains Compounded at Simultaneous Loading on an Axial Tensile Force and Torsion Moment

Determining stress and strain in loaded samples can be accomplished with analytical calculus and numerical or experimental methods [1,2,3,4]. The theoretical determination of stress and strain requires the acceptance of simplified theories on the shape and structure of the element, the mechanical characteristics of the material, and even the loading and support modes [5,6,7]. The material of the element is considered ideal if it is continuous, homogenous, isotropic, and perfectly elastic. The deformation of a straight circular bar subjected simultaneously to tensile and torsion is a complex one, in accordance with the overlap of the linear strain due to its tensile with the angular strain produced as a result of torsion [8,9]. To determine stresses, in this case, the method of effect is considered to be superposition. At tensile and torsion loadings, both the tensile strain and torsion strain are recorded. As a result, the final strain will contain both component linear (ε) and angular (γ). 

Let us consider a straight circular bar made from a continuous, homogenous, isotropic, and perfectly elastic material (Figure 1). The bar is fixed at one end and is free moving at the other, where an axial concentrated force (F) and a concentrated torsion moment (Mt) are applied. As a result of this combined loading, the bar will deform (elastic), along with an element of volume with dx length; the different points of the bar are, after deformation, in areas other than the initial ones. The volume element will initially have angles of 90 degrees at the surface of the bar. After deformation, as a consequence of applying torsion moment, these angles will change with the value γ, which more or less depends on the direction of the application of the torsion moment. On the other hand, the same volume element, under tensile, will elongate in the direction of the bar axis. The transversal sections of the bar do not change their shapes; they just rotate around each other. Considering these observations, the volume element will be in a new position but will suffer deformation, as observed in Figure 1. There will also be an additional decrease in the size of the cross section, which is negligible.

For small deformations, in the elastic domain, we can consider the following:(1)$$\tan(\gamma )\;\approx \;\gamma \;=\;\frac{{R\cdot d\varphi }_{x}}{\mathrm{dx}},$$
where $\frac{{d\varphi }_{x}}{\mathrm{dx}}$ is the specific rotation measured in (rad/m).

We can consider that two points, A and B, located on the bar’s generator after deformation under axial force F and torsion moment Mt, will move in the positions of A’ and B’. Moreover, the volume element we considered will elongate with this value:(2)$$\Delta \mathrm{dx}=\frac{F\cdot \mathrm{dx}}{\mathrm{AE}}=\varepsilon \cdot \mathrm{dx},$$
where A is the area of the transversal section of the bar, E is Young’s modulus, and ε is the tensile strain.

## 2. Comparisons with Other Types of Similar Devices, Description of the Extensometer and Measurement Mode

The extensometer described in this paper is utilized to determine tensile strains ε and torsional strains γ at the same time when a sample is under both tensile and torsional loads. Other types of extensometers designed to simultaneously measure of the axial and torsional strains are known, [9,10,11].

The ones described in [9,10] use four arms to be mounted on the sample, two of which are intended for actual measurement, one for each component of the strain. A feature of both extensometers is that they contain many adjacent pieces. The transmission of the movement resulting from the displacement of the four contact points with the specimen, when the tensile and torsion loading take place, is done by means of levers. Therefore, there is no direct transmission of moving the contact points with the specimen to the sensitive measuring elements. If it doesn’t work very carefully on mounting and setup of the extensometer and align of the contact points with the specimen center, measurement errors may occur. 

At the extensometer described in [10], it can be seen that there are four conductors with electrical signal starting from four measuring elements. In these conditions, it is necessary to have a special software for processing the acquired signals in order to provide, finally, the tensile and torsional strains. With regard to the extensometer described in [11], it is clearly explained that it is, in fact, composed of two different extensometers, one for tensile and the other for torsion, which are contained in a single assembly. Under these conditions, the devices described in [9,10] are quite complicated, with many adjacent components, including zeroing.

The device described in [11], although it has only two arms to be mounted on the sample, contains intermediate parts between the arms and sensors. This can lead to certain errors, if the extensometer is used longer or when it is not handled with care. This extensometer can operate at temperatures up to 1200 °C, which requires certain precautions such as cooling the main elements. At this extensometer, observations could be made in relation to the way of gripping on the sample, which is done by pressing, based on the rigidity (even at high temperature) of the elements adjacent to the two arms that come in contact with the sample.

The extensometer proposed in this paper, Figure 2, has only two arms to be mounted on the sample, one of them being elastic and directly taking the strains. The novelty of this sensor consists in the construction of the elastic arm, with the cross formation of the two elastic beams that will simultaneously measure the two specific deformations. Also, as will be seen, the signals provided by the two elastic elements are not influenced by the complementary loading. Thus, the signal given by the tensile beam is not influenced by the torsion load and the signal provided by the torsion beam is not influenced by the traction load. Compared with the existing extensometers, the one proposed in this paper is more durable, without adjacent components, has a low manufacturing price and, due to its simplicity, has a high accuracy.

The extensometer presented in this paper is intended for simultaneous measurement of the strains, ε and γ, at the tensile and torsion loading. The extensometer with the dimensions of Figure 2 was machined from Al7075. The dimensions of the tensile and torsion elastic beams, 4 and 6, both subject to bending, are designed so that they can measure relatively large specific deformations, up to 6% from axial strains and up to 4.5% from shear strains (torsional strains). With this extensometer, we can perform measurements having a resolution for determining axial strains of 0.112% and for shear strains of 0.105%.

If the extensometer, with the shape of Figure 2, is built with different dimensions and other material, other ranges can be obtained for the axial and shear strains, depending on the desired applications. Obviously, it must ensure the correct elasticity of the tensile and torsion beams and also an adequate rigidity for the other elements. The extensometer can also be used separately for tensile or torsion. The extensometer is mechanically attached to the sample. The two strain transducers, the type instrumented beams with four strain gauges connected in Wheatstone bridges and electrically powered on the diagonal, will provide an electric signal on the other diagonal, which can be transformed, based on a calibration process, into torsional and tensile strains. 

This extensometer is designed to simultaneously retrieve two mechanical strains, transformed by the strain transducers into different electrical signals, which will lead to an exact measurement of the tensile and torsional strains after calibration. The two signals both depend only on the measured process, so they are not influenced by the other strain process. This device can be easily mounted on the samples with just two mounting points and can also be used for cyclical fatigue tensile–torsion loading.

In Figure 2, we present a general image of the extensometer mounted on the sample with the following components: 

1—The rigid supporting basis of the two beams; 2—the rigid bar, with a section approximately constant; 3—the bar containing the elastic measuring beams with cross-shaped manufacturing; 4—the beam used for monitoring tensile strains resulting from the tensile loading(the tensile beam); 5—the four strain gauges (two underneath) mounted on the tensile beam; 6—the beam used for monitoring torsional strains resulting from the torsional loading (the torsion beam); 7—the four strain gauges (two in the back) mounted on the torsional beam; 8—two auxiliary pieces, each with two extensions, required for mounting the extensometer on the sample; 9—two screws with sharp and hardness ends < 30°, screwed in auxiliary pieces 8 and used for mounting the extensometer on the sample; 10—the sample loaded with tensile and torsional strain; 11—elastic rubber bands and / or arcs for mounting the extensometer on the sample.

The extensometer is made out of a rigid bar 1, reported based on the strain resulting from its loading, which supports two bars, the first rigid bar, 2, and the second elastically strained bar, 3, with cut areas 4 and 6. The elastic deformed bar contains two different beams oriented in relation to both loads. There are four strain gauges glued to each elastic beam (5 and 7, also presented in Figure 3).

When the sample is loaded (tensile and torsional loads), both beams are loaded until bending via two different moments: perpendicular to each other and both perpendicular on the longitudinal axis of the arm. The first beam, which is the closest to the mounting tips, is used for measuring tensile strains after tensile loading. The second beam, which is the furthest from these mounting tips, is used for measuring shear (torsional) strains after torsion loading.

The operation of the extensometer is as follows:The sharp tips 9 are mounted on sample 10, which is loaded under a tensile–torsion combined load;The extensometer is mounted on the sample using elastic bands 11 or arcs that pass through the cuttings 8, thereby forcing the easy penetration of the sharp tips through the surface of the sample;Once the loading starts, the signals for the strains are retrieved from the two transducers: the tensile and torsional beams;Using calibration constants, which will be determined according to the steps below, the real strains will be calculated: tensile strain ε and torsional (shear) strain γ.

Due to how the strain gauges are connected in a Wheatstone bridge, both theoretically and practically, each transducer gives a ‘clean’ signal generated by the deformation it monitors. Even though each of the beams is deformed under bending moments on the two planes (Figure 3), the total signal from the strain gauges following the deformation from the ‘residual’ bending moment (aimed at bending the other beam) will be zero. The ‘residual’ bending moment, for each of the two beams, is the one that leads to the deformation reported for the neutral fiber positioned perpendicularly to the minimum dimension.

The electrical binding of the strain gauges on the tensile beam in a Wheatstone bridge was made in order to measure the strain that occurred due to the bending moment, Mz_t_. Thus, the signal retrieved in this case will be:(3)$$\varepsilon_{\mathrm{mas}}=\varepsilon_{t1}+\varepsilon_{t2}-\left(\varepsilon_{c1}+\varepsilon_{c2}\right).$$

Since all strain gauges provide the same signal ε in the modulus, with the compression gauges providing a negative value for the signal, we will have:(4)$$\varepsilon_{\mathrm{mas}}=\varepsilon +\varepsilon -\left(-\varepsilon -\varepsilon \right)=4\varepsilon .$$

We can do the same calculation for the torsion beam (due to the bending moment My_r_).

The bending moment My_r_ results from the torsional loading and is the residual moment for the tensile beam. When the tensile beam is loaded under the bending moment, My_r_ (Figure 3) the strain gauges mounted on the tensile beam deform as follows:The strain gauges m_t1_ and m_c1_ provide the tensile strains (+ε) that occur on the middle part of the beam 5 subjected to tensile due to the residual moment, My_r_;The strain gauges m_t2_ and m_c2_ provide the compression strains (-ε) that occur on the middle part of the beam 5 subjected to compression due to residual moment, My_r_.

The signals provided by the four strain gauges are equal in the module, ε, with those of the compression providing a negative signal. In these conditions, considering the location and connection of the four strain gauges (see Relation (3)), we will have (in the same order as in Relation (3)):(5)$$\varepsilon_{\mathrm{mas}}=\varepsilon -\varepsilon -\left(\varepsilon -\varepsilon \right)=0.$$

As a result, when the tensile beam is loaded with the bending moment My_r_ (residual for the tensile beam), given by the torsion load, the signal of the linear deformation transducer (the tensile beam) will be zero. Similarly, when the torsion beam is loaded with the bending moment Mz_t_ (residual for the torsion beam) given by the tensile load, the signal of the torsion deformation transducer (the torsion beam) will be zero. In this way, each specific deformation transducer will provide signals according to the load for which it was designed: tensile or torsion.

## 3. Calibration for Tensile Loading

The tensile calibration is meant to match the reads made by the gauge strain system (Wheatstone bridge) mounted on the tensile beam, with the real strain obtained by the tensile loading. A ‘control’ sample is used for the tensile loading, and a strain gauge is mounted on it, placed parallel with the longitudinal axis of the sample. This strain gauge mounted on a Wheatstone quarter bridge will provide real tensile strains following the tensile loading of the sample. Both the strain gauge described above and the four strain gauges on the tensile beam will be connected to a system (for example, a Vishay bridge) with power and data acquisition. The control sample will be loaded under tensile stress on a testing machine using movements with small speeds (approximately 1 mm/min).

The data provided below illustrate the determinations made with the extensometer we built with a control sample made out of steel R260Mn (1.0624). Using the data provided by the strain gauge from the control sample under loading and the four strain gauges on the tensile beam (one signal), the graph in Figure 4 was drawn. We observe a good linearity in this graph (considered an approximation line).

Later, by using the extensometer for tensile loading, the signal provided by the tensile strain transducer will need to be multiplied by the calibration coefficient 20.94, thereby revealing the real tensile strain.
Observation: This coefficient is valid for the extensometer we built, with its specific shape, used materials, strain gauge placement, dimensions, etc. For any other extensometer built following this description, a proprietary calibration will be required. The calibration tensile coefficient results are characteristic only for our own extensometer.

To determine the tensile strain when the torsional loading is also performed, the previous coefficient is not sufficient. We need to consider that, along with the elongation Δl of the portion between the supporting points of the extensometer (Figure 5), the cross-sections are rotated one after another.

Assume that we are mounting the extensometer between A and B, where B is located at the end of the bar. If we use the method of effect superposition (valid in the elastic domain), we can say that point A is moving in point A_1_, and point B in B_1_, when the tensile loading takes place. If the torsional loading continues, point A_1_ moves to A’_1_ and point B_1_ to B’_1_. Finally, both initial points (A and B) where the extensometer was mounted will be in A’_1_ and B’_1_ after the tensile–torsion loading.

In Figure 5, we present the following notations: F is tensile force; Mt is torsion moment; φ is the rotation angle of the free-end of the bar; l_0_ is the initial length of the bar; Δl is the total elongation of the bar due to tensile loading; γ is the torsional (shear) strain due to torsional loading; a is the initial distance between the mounting points of the extensometer; Δa is the elongation of the bar in the mounting area of the extensometer (a) if only the tensile stress takes place. When the sample is subjected to only tensile loading, the tensile strain is given by the following relation:(6)$$\varepsilon_{\mathrm{real}}=\frac{{\mathrm{BB}}_{1}}{\mathrm{OB}}=\frac{\Delta l}{l_{0}}=\frac{\Delta a}{a},$$
which is determined based on the total elongation of the bar and the elongation between the mounting points.

When the loading is compounded (tensile and torsion), the signal from the tensile strain transducer, ε_ext_, will be higher than ε_real_ because A’_1_B’_1_ > A_1_B_1_. Under these conditions, the measured signal will contain an error that can be removed based on a computational relationship that will be presented in the following.

The tensile strain provided by the extensometer is:(7)$$\varepsilon_{\mathrm{ext}}=\frac{A_{1}^{\prime }B_{1}^{\prime }-a}{a}=\frac{A_{1}^{\prime }B_{1}^{\prime }}{a}-1,$$
(8)$$A_{1}^{\prime }B_{1}^{\prime }=\frac{a+\Delta a}{\cos\left(\gamma \right)}=\frac{a+\Delta a}{\sqrt{1-\sin^{2}\left(\gamma \right)}}=\frac{a+\Delta a}{\sqrt{1-\gamma^{2}}}\;,$$
considering the small deformation sin(γ) ≈ γ.

When we substitute the last part of Relation (8), which gives us $A_{1}^{\prime }B_{1}^{\prime }$, into Relation (7) while considering Relation (6), we get:(9)$$\varepsilon_{\mathrm{ext}}=\frac{a+\Delta a}{a\sqrt{1-\gamma^{2}}}-1=\frac{\left(1+\varepsilon_{\mathrm{real}}\right)}{\sqrt{1-\gamma^{2}}}-1,$$
which results in:(10)$$\varepsilon_{\mathrm{real}}=\sqrt{1-\gamma^{2}}\left(1+\varepsilon_{\mathrm{ext}}\right)-1,$$
with γ in (m/m).

Therefore, the real tensile strain ε_real_ under simultaneous tensile–torsion loading will be calculated based on the signal given by the tensile strain transducer, ε_ext_, and torsional strain γ, using Relation (10). The torsional strain is determined based on the signal received via the torsion beam, multiplied by a calibration constant, as shown below.

An example of the calculus needed to realistically determine the tensile strain with simultaneous tensile and torsional loading is presented below.

The data obtained by measurements are as follows:ε_ext_ = 600 με = 600·10^−6^ m/m (where the dimension of με is μm/m);γ = 250 με = 250·10^−6^ m/m.

For the tensile strain that is not increased with the calibration coefficient, the following formula is used:$$\varepsilon_{\mathrm{real}}=\sqrt{1-0.00025^{2}}\left(1+0.0006\right)-1\;=\;0.0005999687.$$

The error that results when we do not consider the effect of torsion on tensile strain will be
$$\mathrm{Er}\left[\%\right]=\frac{\varepsilon_{\mathrm{ext}}-\varepsilon_{\mathrm{real}}}{\varepsilon_{\mathrm{real}}}\cdot 100=\frac{600\cdot 10^{-6}-599,968\cdot 10^{-6}}{599,968\cdot 10^{-6}}\cdot 100=0.00521\%.$$

The strains we obtained by measurement and the data presented above were determined for samples made of steel in the elastic domain. Therefore, for these types of strains, the previous calculation leads to very small errors. Consequently, we can consider the results of the tensile strain by directly measuring the tensile beam (multiplied by the calibration coefficient). For higher strain values, especially those resulting from torsion, the tensile strain needs to be determined using Relation (10). After this calculation, multiplication with the calibration constant is required, which provides a value of 20.94 for the extensometer we built.

## 4. Calibration for Torsional Loading

Calibration for torsional loading is provided to match the readings made on the torsion transducer with the real strain resulting from torsion loading. For this purpose, we built a special device that loads a control sample for pure torsional loading. This device is presented in Figure 6 and works as described below.

The control sample instrumented with a T-shaped rosette is fixed in the device. The control sample is mounted on support 16, without the possibility of rotation, using screw 15. Thus, we can consider this sample end as fixed. On the other end of the sample, bearing 5 is present, which is fixed on support 6. Therefore, this end of the sample can rotate on its geometrical axis but cannot move. In this way, the sample is loaded by one torsion moment without other loads. Arm 10 is fixed on this end, as well. When screw nut 11 rotates, arm 10 will rotate around the geometrical axis of the sample, thus creating torsional moment (Mt, Figure 7a). The hole for screw 12 through arm 10 is oriented in the direction of the longitudinal axis of the leverage in order to facilitate the free movement of the leverage.

In order to not insert initial torsional moment due to the weight of arm 10 (and also to allow reversibility of the arm’s movement), a supporting arc was inserted on the bottom side. This arc is able to push towards arm 10 when screw nut 11 is unscrewed. Thus, the measurements can be performed in two ways. In either case, at the beginning of the measurement, the two Wheatstone tensiometric bridges formed on the two sides (strain gauges on the torsion beam and strain gauges on the control sample) are balanced so that the initial signal equals 0, regardless of whether initial loading is present, as this loading is negligible and belongs to the elastic domain.

The calculation for the torsional strain γ is, [12]:(11)$$\gamma =2\varepsilon =\frac{\tau }{G}=\frac{\mathrm{Mt}}{G\cdot \mathrm{Wp}}=\frac{\mathrm{Mt}}{G\cdot \frac{{\pi R}^{3}}{2}}=\frac{2\mathrm{Mt}}{{\pi R}^{3}}\frac{2\left(1+\nu \right)}{E}=\frac{4\mathrm{Mt}\left(1+\nu \right)}{{\pi R}^{3}E}.$$

Calculus Relationship (11) is given just for verification.

The two strain gauges mounted on the control sample are linked in a half bridge, as observed in Figure 7b. The measured signal will be:(12)$$\varepsilon_{\mathrm{mas}}=\varepsilon -\left(-\varepsilon \right)=2\varepsilon ,$$
because one of the samples provides a tensile signal, and the other provides a compression signal.

As a result, the signal measured from the transducer mounted on the control sample will be precisely the torsion strain, γ, according to first part of Relation (11).

Using the data, we acquired from the two strain gauges on the control sample (one signal) and from the four strain gauges on the torsion beam (one signal), the graph in Figure 8 is drawn. The graph has good linearity, and we also show an approximation line. With subsequent uses of the extensometer, the signal provided by the torsion strain transducer will need to be multiplied by a calibration coefficient of 12.77 to obtain the real torsion strain, γ_real_.

Observation: This coefficient is valid for the extensometer we built (i.e., with the specific characteristics, shape and materials, strain gauge positioning, and dimensions. For any other extensometer built as we described, a new calibration needs to be done in order to obtain an appropriate calibration coefficient.

## 5. Conclusions

The extensometer presented in this paper can provide tensile strain and torsional strains at the same time when a sample is loaded under combined torsional and tensile stress. The new characteristic of this extensometer is the manufacture of instrumented beams with four strain gauges each, connected in complete Wheatstone bridges. One of the beams is used for determining tensile strain and the other to determine torsional strain. Considering the way these two elastic beams were made, as well the location of the strain gauges, their residual bending moments do not influence the signals acquired. Thus, each beam will provide a ‘clean’ signal. For each extensometer built as per our description, two different calibrations need to be made: one for tensile strain and one for torsional strain. As we observed, the torsional loading does not significantly influence the measurement of the tensile strain. In these conditions, the signals provided by these elastic beams will only be affected by the calibration constants. 

## Figures and Tables

**Figure 1 sensors-20-00385-f001:**
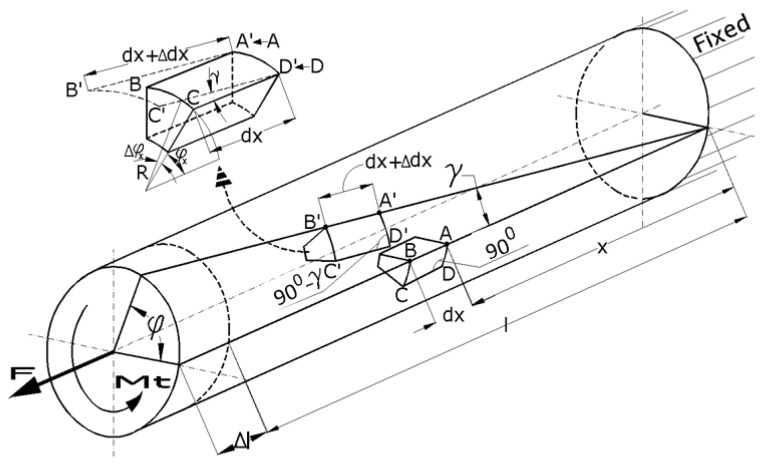
Deformations at the tensile–torsion combined loading.

**Figure 2 sensors-20-00385-f002:**
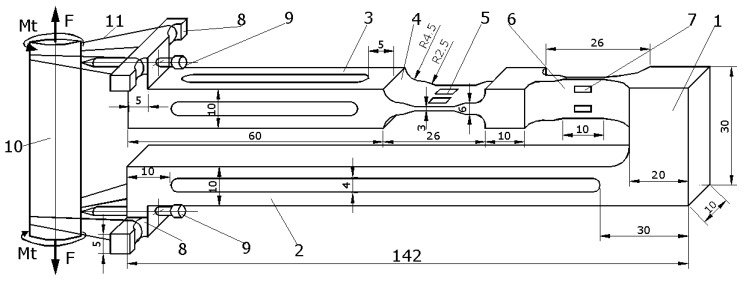
The general image of the extensometer.

**Figure 3 sensors-20-00385-f003:**
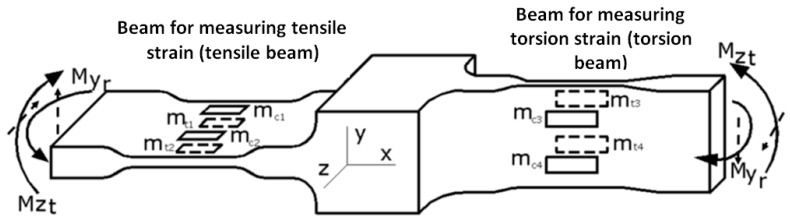
Configuration of the elastic beams.

**Figure 4 sensors-20-00385-f004:**
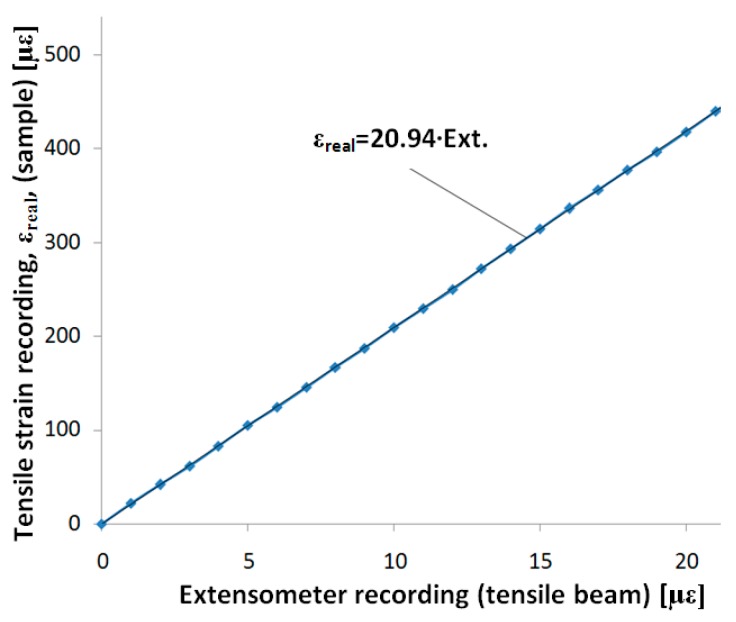
Tensile calibration with the Wheatstone bridge connected to the Vishay bridge P3—the signals from the extensometer and control sample are in με.

**Figure 5 sensors-20-00385-f005:**
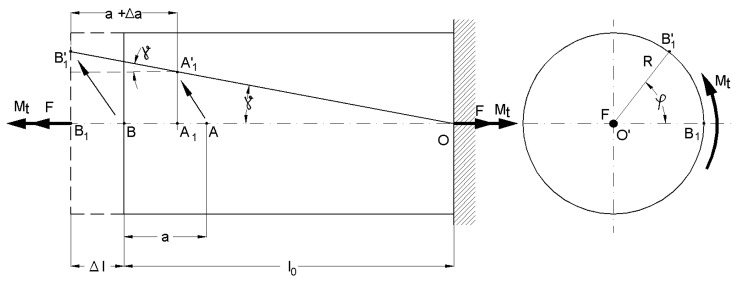
The drawing of the displacement of the fixing points on the sample under tensile–torsion loading.

**Figure 6 sensors-20-00385-f006:**
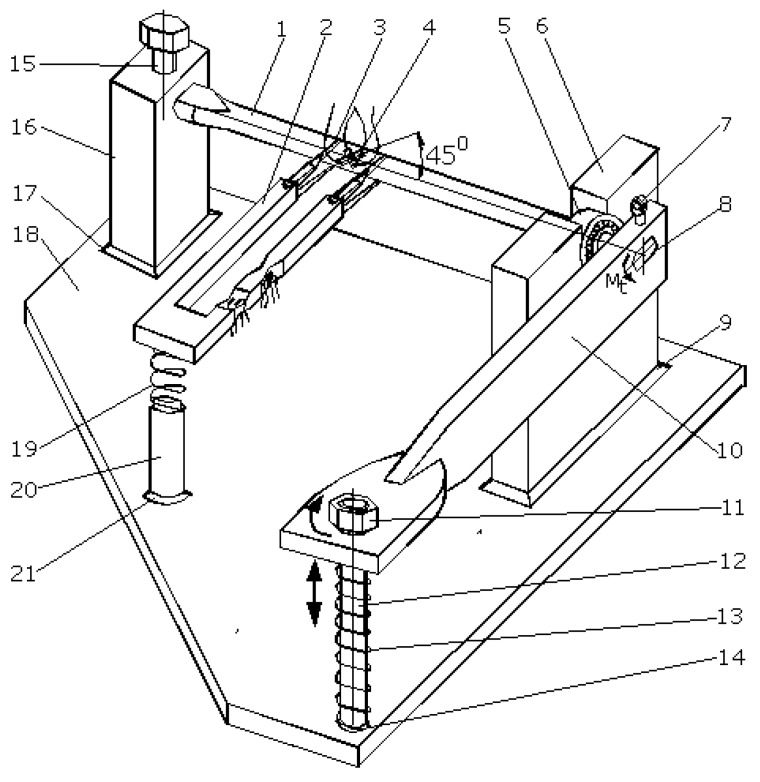
Device for torsional calibration.

**Figure 7 sensors-20-00385-f007:**
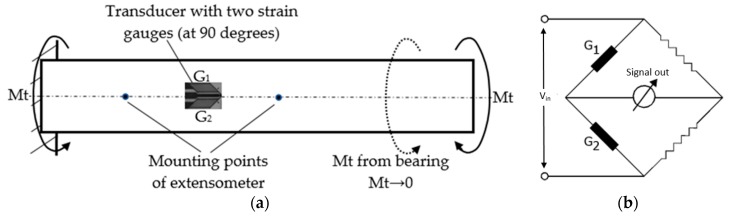
Measuring torsion strain using tensiometric transducers (two strain gauges): (**a**) mounting strain gauges on the control sample; (**b**) linking strain gauges in a half-bridge Wheatstone.

**Figure 8 sensors-20-00385-f008:**
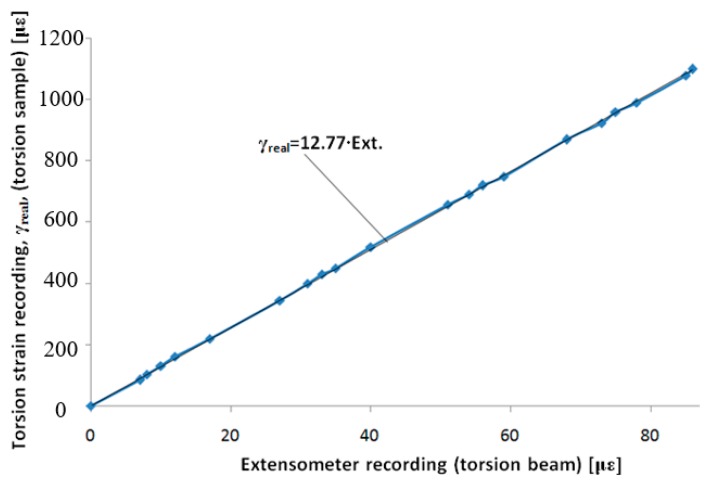
Calibration with all links of the Wheatstone bridge on the Vishay P3 bridge—the signal from the extensometer is in με.

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
