# Peer review of "Extensometer for Determining Strains on a Tensile and Torsion Simultaneous Load"

_sensors, 2020, doi:10.3390/s20020385_

Round 1

Reviewer 1 Report

Dear Author,
the article I had the opportunity to read is, in general, properly structured. The topic is correctly introduced; the working principle is described properly, toghether with the relatede calculation and estimation method for stress-strain estimation.
However, in my experience various devices for the simultaneous measurement of axial and torsion loads are available; therefore, to better understand the innovation provided by the proposed system, my suggestion is to improve the article clarifying the collocation of the new extensometer in comparison with other state-of-the-art solutions. Such description should include:
- the main innovation introduced by the new extensometer (what is new in the device, what is really differentiating it from comparable solutions)
- the research question it aims to respond to (e.g. reduce cost, use less extensimeters, improve precision, improve cycle durability or any other).
I guess that the review part should be also expanded, improving scientific literature research.

Regarding the document presented, let me also suggest a few modifications:
line 32: "final strain that has elements from both loadings": please reformulate the sentence, since I guess it doesn't fit perfect english meaning
line 46: explain why cross section decrease can be considered negligible
line 187-188: 2 equations are numbered as 9
line 199-203: please properly cite eq.9 (which I guess should be eq.10)
Figure 6: I suggest to add another picture showing the constraints scheme, in order to understand why it is supposed to provide a pure torsion solicitation. Has it been considered the possibility of a non-ideal functionality of the bearing? is the machine already validated or has it been built for this application? is it possible to provided a (brief) literature review on main schemes usually adopted for pure-torsion testing machines?

Author Response

Point 0:

Unfortunately I am not allowed to upload the final paper at this stage. This is in the attached file as the complete answer (with figures) at the end of the paper.

These answers do not contain figures. See in the attached at the end of the paper the complete answer.

Modifications in the new version of the work:

- red areas in the paper;

- Figure 7;

- small changes in format, text references, final references, etc.

Dear Author,
the article I had the opportunity to read is, in general, properly structured. The topic is correctly introduced; the working principle is described properly, toghether with the relatede calculation and estimation method for stress-strain estimation.

Point 1:
“various devices for the simultaneous measurement of axial and torsion loads are available; therefore, to better understand the innovation provided by the proposed system, my suggestion is to improve the article clarifying the collocation of the new extensometer in comparison with other state-of-the-art solutions. Such description should include:
- the main innovation introduced by the new extensometer (what is new in the device, what is really differentiating it from comparable solutions)
- the research question it aims to respond to (e.g. reduce cost, use less extensimeters, improve precision, improve cycle durability or any other).
I guess that the review part should be also expanded, improving scientific literature research.”

Response 1:

This part of the answer (with italic) is included in the paper between rows 61-73, with. Images with the three types of extensometers cited in [9,10,11] are shown below.

Other types of extensometers designed to simultaneously measure the axial deflection and torsional angle-of-twist are known [9,10,11]. The ones described in [9,10] use four arms to be mounted on the sample, two of which are intended for actual measurement, one for each component of the strain. Under these conditions, the devices are quite complicated, with many adjacent components, including zeroing. The last device [11], although it has only two arms to be mounted on the sample, contains intermediate parts between the arms and sensors. This can lead to certain errors, if the extensometer is used longer or when it is not handled with care.

The sensor proposed in this paper has only two arms to be mounted on the sample, one of them being elastic and directly taking the strains. The novelty of this sensor consists in the construction of the elastic arm, with the cross formation of the two elastic beams that will simultaneously measure the two specific deformations. Compared with the existing extensometers, the one proposed in this paper is more durable, without adjacent components, has a low manufacturing price and, due to its simplicity, has a high accuracy.

Point 2:

Regarding the document presented, let me also suggest a few modifications:
line 32: "final strain that has elements from both loadings": please reformulate the sentence, since I guess it doesn't fit perfect english meaning

Response 2:

I modified in the new version of the paper. Now it's on line 31 and 32.

Point 3:

line 46: explain why cross section decrease can be considered negligible

Response3:

Under the action of the traction load, the area of the cross section is decreases. At elastic stresses, such as those for which the extensometer is intended, the cross-section decrease is very small relative to the total area.

Point 4:

line 187-188: 2 equations are numbered as 9

Response 4:

I modified in the new version of the paper, as well as the following references and their inclusion in the text - now on line 196.

Point 5:

line 199-203: please properly cite eq.9 (which I guess should be eq.10)

Response 5:

I modified in the new version of the paper

Point 6:

Figure 6: I suggest to add another picture showing the constraints scheme, in order to understand why it is supposed to provide a pure torsion solicitation. Has it been considered the possibility of a non-ideal functionality of the bearing?

 is the machine already validated or has it been built for this application? is it possible to provided a (brief) literature review on main schemes usually adopted for pure-torsion testing machines?

Response 6:

I have restored figure 7 which contains your requirement relative to constraint from figure 6 – see file attached.

This device was created by us, to introduce the pure torsion on the specimen on which a strain gauge transducer (two strain gauges at 90 degrees) is mounted which provides the torsion strain. The load and support scheme is now introduced in Figure 7a. There are both devices and special machines for the pure torsion load. For the simplicity of the test and the determinations we made the one presented in the paper. Through this device it was important not to introduce normal stresses into the sample at the cross section of the specimen, either by traction or by bending. This is followed by the device's design. For calibration, the direct connection is made between the signal supplied by the strain gauge transducer mounted on the sample and the singular signal provided by the four strain gauges on the beam intended for the torsion signal of the extensometer. The torque applied does not take into account in the calibration, the last part of the relation (11) (after renumbering) being for verification only. As a result, even if there is a reaction torque in the bearing (very small due to friction) it is not related to the calibration.We do not have a testing machine to loading in torsion, so we made this device.

See in the attached file.

(a)                                                                                                (b)

Figure 7. Measuring torsion strain using tensiometric transducers (two strain gauges).

See in the attached file

Tensile calibration - images

See in the attached file

Images during torsion calibration

Reviewer 2 Report

Please structure the paper as required in the format according to journal. Please provide verification to show the validity of the measurement under combined tension and torsion using direct measurement with strain gauges or other sensing elements on the test specimen. In doing (2), you need to conduct the identical experiment with and without the proposed extensometer. You need to show that the extensometer does not impact on the measurement. 

The author needs to:

a) Verify the results using strain gauges bonded onto the specimen;

b) Without what are the results like with and without the extensometer to ensure that the "stiffness" of the extensometer does not affect the strain reading.

Author Response

Point 0:

Unfortunately I am not allowed to upload the final paper at this stage. This is in the attached file as the complete answer (with figures) at the end of the paper.

These answers do not contain figures. See in the attached at the end of the paper the complete answer.

Modifications in the new version of the work:

- red areas in the paper, italic areas below;

- Figure 7;

- small changes in format, text references, final references, etc.

This part of the answer (with red) is included in the paper between rows 61-73, with. Images with the three types of extensometers cited in [9,10,11] are shown at the end of the att. file – is not included in the final paper.

Other types of extensometers designed to simultaneously measure the axial deflection and torsional angle-of-twist are known [9,10,11]. The ones described in [9,10] use four arms to be mounted on the sample, two of which are intended for actual measurement, one for each component of the strain. Under these conditions, the devices are quite complicated, with many adjacent components, including zeroing. The last device [11], although it has only two arms to be mounted on the sample, contains intermediate parts between the arms and sensors. This can lead to certain errors, if the extensometer is used longer or when it is not handled with care.

The sensor proposed in this paper has only two arms to be mounted on the sample, one of them being elastic and directly taking the strains. The novelty of this sensor consists in the construction of the elastic arm, with the cross formation of the two elastic beams that will simultaneously measure the two specific deformations. Compared with the existing extensometers, the one proposed in this paper is more durable, without adjacent components, has a low manufacturing price and, due to its simplicity, has a high accuracy.

See in the attached file

(a)                                                    (b)

Figure 7. Measuring torsion strain using tensiometric transducers (two strain gauges).

Point 1:

Please structure the paper as required in the format according to journal.

Response 1:

I reviewed the entire paper and also instructions for authors and redone where it was necessary.

Modifications in the new version of the work:

- red areas;

- Figure 7;

- changes in format, text references, final references, etc.

Point 2:

Please provide verification to show the validity of the measurement under combined tension and torsion using direct measurement with strain gauges or other sensing elements on the test specimen. In doing (2), you need to conduct the identical experiment with and without the proposed extensometer. You need to show that the extensometer does not impact on the measurement. 

Response 2:

We used the traction and torsion tests separately: the first on a universal testing machine type Instron 8801 and the second on the torsion device created by us – see images below.

Unfortunately, we do not have a testing machine to make the simultaneous loading for traction with torsion. However, the results obtained by the calibration tests separated by traction and torsion are correct.

This device from torsion was created by us, to introduce the pure torsion on the specimen on which a strain gauge transducer (two strain gauges at 90 degrees) is mounted which provides the torsion strain. The load and support scheme is now introduced in Figure 7a. There are both devices and special machines for the pure torsion load. For the simplicity of the test and the determinations we made the one presented in the paper. Through this device it was important not to introduce normal stresses into the sample at the cross section of the specimen, either by traction or by bending. This is followed by the device's design. For calibration, the direct connection is made between the signal supplied by the strain gauge transducer mounted on the sample and the singular signal provided by the four strain gauges on the beam intended for the torsion signal of the extensometer. The torque applied does not take into account in the calibration, the last part of the relation (11) (after renumbering) being for verification only. As a result, even if there is a reaction torque in the bearing (very small due to friction) it is not related to the calibration.

See in the attached file

Tensile calibration - images

See in the attached file

Images during torsion calibration

Point 3:

You need to show that the extensometer does not impact on the measurement. 

Response 3:

The calibration of the extensometer was performed by matching the specific deformations on the samples required for traction and torsion (taken from tensometric transducers) with the signal provided by each of the two Wheatsone bridges mounted on the extensometer. The signal taken from the transducers (strain gauges - linear and torsional) of the samples subjected to traction and torsion cannot be questioned. By calibration, from each of the Wheatsone bridges, the real values of the deformations from the traction components and the torsion component are obtained. As a result, the construction of the extensometer does not influence the signal provided by the two Wheatstonebridges.

The author needs to:

Point 4:

Verify the results using strain gauges bonded onto the specimen;

Response 4:

This was done through calibration tests, separately for each load. It is true that we cannot perform a simultaneous test on traction and torsion because we do not have such a testing machine available. However, the calibration constants remain valid and the results that will be obtained by simultaneous torsion-traction test will be the real ones, with the observation resulting from the relation (10) and the calculated error.

See in the attached file

Strain gauges on the sample for torsion               Strain gauges on the sample for tensile

Point 5:

Without what are the results like with and without the extensometer to ensure that the "stiffness" of the extensometer does not affect the strain reading.

Response 5:

The extensometer can be constructed from different materials and to different sizes. The one built by us and which can be seen in the image below, is made from Al7075. The dimensions of the bending and torsion elastic beams are designed so that they can measure relatively large specific deformations. Instead, this type of extensometer can be constructed from various materials and with the elasticity of the measuring beams and the rigidity of the other desired beams. Appropriate rigidity must be ensured, both for the parallel bars 2 and 3 and for the rigid bar 1 in relation to the elastic deformation zones 5 and 7. This remains at the discretion of the one who will build his extensometer according to the sketch in figure below.

Reviewer 3 Report

This paper describes the extensometer that can provide tensile and torsional strains at the 
same time. Although such extensometer is already available, for example, as a product of MTS, the design is new and this paper may be interesting to readers of Sensors. However, I noticed the following points that are not described in the current version of this manuscript. I think this paper will be more interesting one if these are treated in a revised edition.

Measurement ranges of the extensometer. Although the measurement ranges may depend on various dimensions of the extensometer, this information may be useful for readers. For example, limitations of the linear relationships shown in Figs. 4 and 8 are valuable information to readers. Rigidity of the extensometer. When the rigidity of the extensometer is too high, it affects elastic deformation behavior of specimens. On the other hand, if it is too low, handling of the extensometer may be difficult. The appropriate rigidity seems to be a function of elastic constants of specimens. I think this information is also interesting to readers.

Author Response

Response 0:

Unfortunately I am not allowed to upload the final paper at this stage. This is in the attached file as the complete answer (with figures) at the end of the paper.

These answers do not contain figures. See in the attached at the end of the paper the complete answer.

Modifications in the new version of the work:

- red areas in the paper, italic areas below;;

- Figure 7;

- small changes in format, text references, final references, etc.

This part of the answer (with red) is included in the paper between rows 61-73, with. Images with the three types of extensometers cited in [9,10,11] are shown at the end of tne paper attached – is not included in the final paper.

Other types of extensometers designed to simultaneously measure the axial deflection and torsional angle-of-twist are known [9,10,11]. The ones described in [9,10] use four arms to be mounted on the sample, two of which are intended for actual measurement, one for each component of the strain. Under these conditions, the devices are quite complicated, with many adjacent components, including zeroing. The last device [11], although it has only two arms to be mounted on the sample, contains intermediate parts between the arms and sensors. This can lead to certain errors, if the extensometer is used longer or when it is not handled with care.

The sensor proposed in this paper has only two arms to be mounted on the sample, one of them being elastic and directly taking the strains. The novelty of this sensor consists in the construction of the elastic arm, with the cross formation of the two elastic beams that will simultaneously measure the two specific deformations. Compared with the existing extensometers, the one proposed in this paper is more durable, without adjacent components, has a low manufacturing price and, due to its simplicity, has a high accuracy.

Point 1:

Measurement ranges of the extensometer. Although the measurement ranges may depend on various dimensions of the extensometer, this information may be useful for readers. For example, limitations of the linear relationships shown in Figs. 4 and 8 are valuable information to readers.

Response 1:

The extensometer can be constructed from different materials and to different sizes. The one built by us and which can be seen in the image below, is made from Al7075. The dimensions of the bending and torsion elastic beams are designed so that they can measure relatively large specific deformations. Instead, this type of extensometer can be constructed from various materials and with the elasticity of the measuring beams and the rigidity of the other desired beams. Appropriate rigidity must be ensured, both for the parallel bars 2 and 3 and for the rigid bar 1 in relation to the elastic deformation zones 5 and 7. This remains at the discretion of the one who will build his extensometer according to the sketch in figure below.

Point 2:

Rigidity of the extensometer. When the rigidity of the extensometer is too high, it affects elastic deformation behavior of specimens. On the other hand, if it is too low, handling of the extensometer may be difficult. The appropriate rigidity seems to be a function of elastic constants of specimens. I think this information is also interesting to readers.

Response 2:

As shown above and as can be seen in figure 1, we did not indicate the dimensions nor the material from which the extensometer can be made. As a result, each one builds it from the material and dimensions to ensure both the rigidity of the corresponding elements and the desired elasticity of the elastic beams. Our extensometer was built from Al7075, with the dimensions below. The 3 mm dimension of the elastic beams (5 and 7 in figure 1) was obtained on the basis of several attempts at initially larger dimensions. It has reached the size of 3 mm considering the strains area we use, for ductile steel and aluminum. I cannot refer to the report the rigidity in the paper because there are no dimensions. However, for the extensometer in the figure below we have:

As a result, for the extensometer built by us, the rigidity of bars 2 and 3 is 33 times higher than the elastic beams 5 and 7.

See in the attached file

Tensile calibration - images

See in the attached file

Images during torsion calibration

Round 2

Reviewer 1 Report

Dear Authors, 

thanks for providing your review.

Author Response

Thank you for your review comments to help us improve the quality of our manuscript.